
# Cloud Detection from Multi-Angular Polarimetric Satellite Measurements using a Neural Network Ensemble Approach

Zihao Yuan[1,2], Guangliang Fu[1], Bastiaan van Diedenhoven[1], Hai Xiang Lin[2,3], Jan Willem Erisman[2], and Otto P. Hasekamp[1]

[1]Netherlands Institute for Space Research (SRON, NWO-I), Leiden, the Netherlands
[2]Institute of Environmental Science (CML), Leiden University, Leiden, the Netherlands
[3]Delft Institute of Applied Mathematics, Delft University of Technology, Delft, the Netherlands

**Correspondence:** Z.Yuan (z.yuan@sron.nl)

**Abstract.** This paper describes a neural network cloud masking scheme from PARASOL (Polarisation and Anisotropy of Reflectances for Atmospheric Science coupled with Observations from a Lidar) Multi-Angle Polarimetric measurements. The algorithm has been trained on synthetic measurements and has been applied to the processing of one year of PARASOL data. Comparisons of the retrieved cloud fraction with MODIS (Moderate Resolution Imaging Spectroradiometer) products show

overall agreement in spatial and temporal patterns but the PARASOL-NN retrieves lower cloud fractions. Comparisons with a goodness-of-fit mask from aerosol retrievals suggest that the NN cloud mask flags less clear pixels as cloudy than MODIS ($\sim$ 3% of the clear pixels, versus $\sim$ 15% by MODIS). On the other hand the NN classifies more pixels incorrectly as clear than MODIS ($\sim$ 19% by NN, versus $\sim$ 15% by MODIS). Additionally, the NN and MODIS cloud mask have been applied to the aerosol retrievals from PARASOL using the Remote Sensing of Trace Gas and Aerosol Products (RemoTAP) algorithm.

Validation with AERONET shows that the NN cloud mask performs comparably with MODIS in screening residual cloud contamination in retrieved aerosol properties. Our study demonstrates that cloud masking from MAP aerosol retrievals can be performed based on the MAP measurements themselves, making the retrievals independent of the availability of a cloud imager.

## 1 Introduction

Climate change, which refers to long-term changes in temperature and weather patterns, has received large public attention and is a major global concern. An important limitation in our ability to understand, quantify, and predict climate change is related to the roles of aerosols and clouds in our climate system (Bellouin et al., 2020). Anthropogenic aerosols have the potential to cause a radiative forcing that is comparable in magnitude to greenhouse gases but with an opposite sign. While the climate impact of greenhouse gases is relatively well-understood, the cooling effect (negative forcing) caused by aerosols

is the largest source of uncertainty in the latest Intergovernmental Panel on Climate Change (IPCC) assessment, as well as in previous assessments (Arias et al., 2021).

Aerosols affect Earth's climate by scattering and absorbing radiation (aerosol-radiation interactions) and acting as condensation nuclei for cloud droplets and ice crystals (aerosol-cloud interactions). Substantial changes in anthropogenic aerosol





emissions occurred in the industrial era, as the result of fossil fuel burning in transport, industry, and energy production. The
most uncertain aspects in aerosol radiative forcing are related to aerosol-cloud interactions (Bellouin et al., 2020). By acting
as Cloud Condensation Nuclei (CCN), aerosols affect the cloud droplet number concentration ($N_d$) and consequently, the
cloud albedo, causing the Radiative Forcing due to aerosol-cloud interaction (RFaci) (Twomey, 1974). Subsequently, rapid
adjustments take place in e.g. Cloud Fraction (CF) and Liquid Water Path (LWP) that result from an initial change in $N_d$.
The combination of RFaci and adjustments results in Effective Radiative Forcing due to aerosol-cloud interactions (ERFaci).
Apart from their interactions with aerosols, also clouds alone represent an important uncertainty in future climate predictions.
Specifically, the impact of changing temperatures on clouds, known as cloud feedback, remains poorly understood (Zelinka
et al., 2022).

Satellite-based remote sensing provides essential information for understanding and quantification of aerosol-radiation in-
teractions (Myhre et al., 2009; Lacagnina et al., 2015, 2017; Chen et al., 2022) and aerosol-cloud interactions (Gryspeerdt
et al., 2017; Hasekamp et al., 2019b; Quaas et al., 2020; Gryspeerdt et al., 2023). For aerosols, the richest set of information
on aerosol properties can be obtained from instruments that measure both intensity and polarization of scattered sunlight at
multiple wavelengths and multiple viewing angles for one ground pixel (Mishchenko and Travis, 1997; Hasekamp and Land-
graf, 2007; Dubovik et al., 2019). In this paper, we refer to such an instrument as a Multi-Angle Polarimeter (MAP). The
POLarization and Directionality of Earth Reflectances (POLDER) instrument has flown in three different incarnations and has
pioneered space-based MAP remote sensing. Until now, the only instrument of this type that has provided a multi-year data
record is the POLDER-3 instrument on the PARASOL (Polarisation and Anisotropy of Reflectances for Atmospheric Science
coupled with Observations from a Lidar) micro-satellite, in orbit between 2004-2013. The successor mission of POLDER is
the 3MI (Fougnie et al., 2018) on ESA/EUMETSAT Metop SG-A satellite. The instrument suite of the NASA Phytoplank-
ton, Aerosol, Cloud, and ocean Ecosystem (PACE) mission (Werdell et al., 2019), to be launched in January 2024, will have
the capability to improve substantially on both aerosol and cloud retrievals by performing multi-angle polarimetric (MAP)
measurements at higher accuracy, more wavelengths (hyper-spectral), and more viewing angles (hyper-angular) than exist-
ing instruments. PACE will carry two polarimeters: i) The SPEXone instrument (Hasekamp et al., 2019a), contributed by the
Netherlands, will provide new and improved measurements of aerosol microphysical properties (size distribution, refractive
index, shape) and optical properties (Aerosol Optical Depth-AOD, and Single Scattering Albedo-SSA). ii) The Hyper-Angular
Rainbow Polarimeter-2 (HARP-2) has the capability to provide improved cloud properties ($N_d$, droplet size distribution, cloud
phase) from the cloud bow in polarization (Grosvenor et al., 2018). The PACE mission will be the first to provide polarimetric
retrievals of aerosol and cloud properties in more than a decade.

Currently, aerosol properties from these instruments can only be retrieved in cloud-free areas, or for areas where an aerosol
is located above a cloud (Waquet et al., 2009, 2013; Knobelspiesse et al., 2015). Therefore, it is important to be able to identify
cloud-free pixels (cloud screening) for which an aerosol retrieval algorithm can provide meaningful output. Stap et al. (2014)
have shown that the goodness-of-fit of aerosol retrieval from MAP measurements can be used for cloud screening. The idea
behind this, is that for a measurement that is contaminated by a cloud, the forward model of the aerosol retrieval cannot find
a good fit with the measurement. However, a disadvantage of this approach is that it requires large computation time because





the aerosol retrieval has to be applied on all pixels (cloudy and cloud-free) and the cloud-free pixels can only be identified

after the retrieval. Here it is important to note that the aerosol retrieval procedure itself is computationally expensive because it needs online radiative transfer calculations in order to provide a sufficiently accurate result. Cloudy pixels would even require larger computation time because of the large optical thickness. Given that the results for the cloudy pixels (∼80% of all pixels) are discarded, processing of these cloudy pixels can be considered as redundant computation time. To avoid this redundant computation, cloud masking should be applied before the retrieval.

The goal of this work is to develop a cloud screening procedure for aerosol retrievals from MAP instruments. Here, we focus on the POLDER-3/PARASOL instrument (hereafter simply referred to as PARASOL) as it is the only MAP that has provided a multi-year data set. Currently, the most accurate cloud mask for aerosol retrievals from PARASOL is provided by MODIS (Moderate-Resolution Imaging Spectroradiometer). The MODIS cloud mask (Ackerman et al., 1998), which provides measurements within three minutes from PARASOL (for the period PARASOL was a part of the NASA A-Train), is based on

input signals from visible and infrared bands, which detect the high, spectrally flat reflectance and low brightness temperature feature of clouds. Although the cloud mask from MODIS has been shown to be useful for performing cloud screening for PARASOL aerosol retrievals, there are two important reasons to develop a cloud screening algorithm based on PARASOL alone: First of all, not all PARASOL pixels have a corresponding MODIS cloud retrieval. Only for the period 2005-2009, PARASOL and MODIS measurements were co-located in time and location because both instruments flew in the same or-

bit (NASA A-Train). So, for the period 2010-2013, no MODIS measurements are available to perform cloud screening for PARASOL. Also for 2005-2009, there are sometimes orbits missing in the MODIS data because of instrument switch-offs or processing problems. Another motivation for developing a PARASOL-only cloud mask is that the MAP measurements of PARASOL contain unique sensitivity to clouds and this may result in better capability to screen for clouds than MODIS, e.g., in the case of dust aerosols (Wang et al., 2016).

Cloud fraction retrievals from PARASOL were performed operationally at $18 \times 18$ km$^2$ (Zeng et al., 2011). These cloud fraction retrievals are based on thresholds on measured radiances, polarized radiances, and apparent oxygen pressure at $6 \times 6$km$^2$. To avoid the choice of priori thresholds which define whether a pixel is cloud contaminated or not, we develop a retrieval of cloud fraction (CF) and use the retrieved cloud fraction to define one or more cloud masks (e.g., a strict and loose one). The cloud mask is at the native spatial resolution of PARASOL which is $\sim 6 \times 6$ km$^2$. Given that the MAP measurements

are also affected by aerosol properties and surface reflection properties, a classical retrieval algorithm (by radiative transfer calculations and inversion) for the retrieval of CF would be very complex because it would need, in addition to CF, to include many fit parameters related to aerosol (e.g., amount, size, refractive index, layer height) and surface Bi-directional Reflection Distribution Function (BRDF) parameters. Apart from the fact that such a retrieval algorithm requires a prohibitive amount of computing power, it may also have the risk of not finding an optimal solution (e.g., by ending at a local minimum of the

inversion cost function). The use of Neural Networks (NNs) is a promising way because of the efficiency in computation and the possibility to define a reduced state vector (with just CF), while still taking into account the effect of aerosol and surface reflection. NNs have been used successfully in polarimetric remote sensing of aerosols by e.g. Di Noia et al. (2017), and Gao et al. (2021), as well as for polarimetric remote sensing of cloud microphysical properties by Di Noia et al. (2019).



The paper is organized as follows: Section 2 introduces the data used in the study, Section 3 describes the training method-
ology of the Neural Network with a consistency check based on a synthetic evaluation set, Section 4 shows the data processing
of one year (2008) PARASOL measurements and a comparison with MODIS, a goodness-of-fit mask from RemoTAP aerosol
retrieval, and AERONET data. Finally, Section 5 summarizes and concludes this study.

## 2 Data description

### 2.1 PARASOL

PARASOL was operational from 2004 to 2013, being part of the NASA A-Train satellite constellation in synthesis with
MODIS/AQUA (multi-spectral imager), CALIPSO (lidar), and CLOUDSAT until 2009. PARASOL has provided intensity
measurements in 9 spectral bands (443, 490, 565, 670, 763, 765, 865, 910, 1020 nm) and linear polarization (Stokes parameters
Q and U) in 3 spectral bands (490, 670, 865 nm), viewing a ground scene under (up to) 16 different viewing geometries (Fougnie
et al., 2007). The level 1 (non-cloud-screened) observations are available on a sinusoidally projected grid of $\sim 6 \times 6$ km$^2$ pixels,
named the full-resolution (FR) grid. In this work, we only use PARASOL measurements within the latitude ranges from 60° S
to 60° N that have at least 14 viewing angles available.

### 2.2 PARASOL RemoTAP aerosol retrievals

The Remote sensing of Trace gas and Aerosol Products (RemoTAP) algorithm (Hasekamp et al., 2011; Fu and Hasekamp,
2018; Fu et al., 2020; Lu et al., 2022) is a flexible algorithm that can be used for retrieval of optical and microphysical aerosol
properties from Multi-Angle Polarimeter (MAP) measurements, retrieval of aerosol properties and trace gas columns (e.g.
carbon dioxide (CO$_2$) and methane (CH$_4$)) from spectroscopic measurements, or for joint retrievals using multiple instruments
(e.g., MAP and spectrometer together). RemoTAP is based on iteratively fitting a linearized radiative transfer model (Hasekamp
and Landgraf, 2002, 2005; Schepers et al., 2014) to the measurements of intensity and polarization of light reflected by the
Earth's atmosphere and surface. It has large flexibility in the definition of parameters to be retrieved and allows retrievals over
land, ocean, and clouds.

For the setup of the aerosol retrieval, we utilize a parametric 3-mode aerosol description, where the size distribution is
characterized by 3 log-normal modes (i.e., $N_{\mathrm{modes}} = 3$), comprising one fine mode and two coarse modes (dust and soluble).
The state vector for the fine mode includes parameters such as effective radius $r_{\mathrm{eff}}$, effective variance $v_{\mathrm{eff}}$, aerosol column
number $N_{\mathrm{aer}}$, spherical fraction $f_{\mathrm{sph}}$, and refractive index coefficients $c_{\mathrm{k}}$, which correspond to the standard refractive index
spectra of inorganic aerosol (real part), black carbon (imaginary part), organic carbon (imaginary part), and water. The dust
mode consists of non-spherical dust, and its state vector includes $r_{\mathrm{eff}}$, $N_{\mathrm{aer}}$, and a coefficient for the imaginary part of the dust
refractive index. The fixed parameters are $f_{\mathrm{sph}} = 0$, $v_{\mathrm{eff}} = 0.6$, and $c_{\mathrm{k}} = 1$ for the real part of the dust refractive index. The third
mode represents a coarse soluble mode, and its state vector includes $r_{\mathrm{eff}}$, $N_{\mathrm{aer}}$, and a coefficient $c_{\mathrm{k}}$ of the inorganic refractive





index spectrum. The fixed parameters for this mode are $f_{sph} = 1$, and $v_{eff} = 0.6$. A detailed description can be found in Lu et al.
125     (2022).

## 2.3    Cloud fraction from MODIS-Aqua cloud mask product

The MODIS Cloud Mask product used in this work is a Level 2 product generated at 1-km (at nadir) spatial resolutions from
MODIS-Aqua. The algorithm, as described in Ackerman et al. (1998), employs a series of visible and infrared threshold and
consistency tests to specify the confidence that an unobstructed view of the Earth's surface is observed. An indication of
shadows affecting the scene is also provided. There are 4 cloud flag categories in the product: confidently cloudy, uncertain
cloudy, probably clear, and definitely clear. The cloud fraction (referred to as MODIS cloud fraction hereafter) is calculated as
the fraction of confidently- and uncertain-cloudy-flagged 1-km-resolution MODIS pixels within a $6\text{km} \times 6\text{km}$ PARASOL grid.

## 2.4    AERONET Data

The RemoTAP aerosol retrievals are validated by data from the Aerosol Robotic Network (AERONET, Holben et al. (2001)).
The aerosol optical depth (AOD) at 550nm is from AERONET level 2 data (version 3.0). The Single scattering albedo (SSA)
at 550nm is from AERONET level 2 Almucantar retrieval inversion products (Dubovik and King, 2000; Dubovik et al.,
2000, 2002).

## 3    Neural network design

### 3.1    Training set generation

Our neural network approach for the retrieval of cloud fraction from PARASOL data was decomposed into 2 subtasks, with
different neural networks dedicated to pixels over land and ocean, respectively.

      The training of the NNs in this work uses synthetic measurements created with the radiative transfer model (RTM): LIN-
TRAN (Schepers et al., 2014). In our setup, LINTRAN computes the top-of-atmosphere intensity vector $\mathbf{I}$ with Stokes param-
eters $I$, $Q$, $U$ as components as a function of wavelength and viewing-solar geometry. In the simulation process, the ocean
reflection properties are parameterized with wind speed as in Cox and Munk (1954), the chlorophyll-a concentration as in
Chowdhary et al. (2006) and Fan et al. (2019). For the simulations over land, the surface bidirectional reflectance distribution
function (BRDF) is parameterized using the Ross-Li model (Wanner et al., 1995), and the bidirectional polarization distribu-
tion function (BPDF) is parameterized as in Maignan et al. (2009). Liquid clouds are described by spherical particles with a
Gamma size distribution, with the refractive index of water taken from Hess et al. (1998). For ice clouds, hexagonal crystals
with varying aspect ratios and surface distortions are used as proxies for complex crystals (van Diedenhoven et al., 2020). The
aerosol size distribution is based on three log-normal modes as Lu et al. (2022) where each mode is described by the effective
radius, effective variance, complex refractive index (wavelength dependent), aerosol optical depth (AOD) at 550nm, fraction





of spherical particles, aerosol layer height. The aerosols are considered homogeneously distributed between 0-1 km in our simulations.

The distribution of the different input parameters for the training set is described in Table 1. The cloud properties are taken from a random distribution. To roughly represent the true cloud fraction distribution and also to emphasize small cloud fractions (which matters for cloud masking), 40% of cloud fractions are generated between 0.2 and 1, 20% are between 0 and 0.2, and 20% completely clear and 20% completely cloudy pixels are also included in the data set. The aerosol and surface (land and ocean) properties are from randomly picked pixels of RemoTAP global retrieval for the year 2008. The combination of solar

zenith angles, viewing zenith angles, and relative azimuth angles are randomly picked from PARASOL level-1 measurements for the year 2008. Here it should be noted that only the measurements performed at a minimum of 14 angles are considered for training the neural networks. The choice was made in order to avoid dealing with the difficulty of having an input vector of variable size or, as an alternative, of passing input vectors with missing data to the neural networks.

The NN was designed to produce cloud fraction (CF) as output, so the partially cloudy measurements and "true" cloud

fraction are needed in the training process. To model the intensity vector $\mathbf{I}$ for a partially cloudy pixel with cloud fraction $f$, the independent pixel approximation (IPA) was used:

$$\mathbf{I}_{\text{ipa}} = f\,\mathbf{I}_{\text{cloudy}} + (1-f)\,\mathbf{I}_{\text{clear}}, \tag{1}$$

where the vectors $\mathbf{I}_{\text{cloudy}}$ and $\mathbf{I}_{\text{clear}}$ correspond to simulations for cloudy conditions (either fully covered by liquid cloud or ice cloud) and clear conditions. Cloud screening task for aerosol retrieval pays more attention to small cloud fractions, where a

larger sensitivity is required. Therefore, instead of feeding the cloud fraction directly into the NN, we apply the inversion of the Sigmoid function:

$$T = \ln(\frac{f}{1-f}), \tag{2}$$

where $T$ is the training target fed in the NN, and $f$ is the original cloud fraction. Here it should also be noticed that $f$ is clipped between $10^{-5}$ and $(1-10^{-5})$, as the function $T$ has no definition at $f=0$ or $f=1$. We have investigated also other functions

for T (logarithmic, linear, step functions), but the inverse Sigmoid provides the best results for cloud screening.

Based on the same set of aerosol and surface properties, synthetic forward calculations for three scenes should be generated: fully clear, fully covered by liquid cloud, and fully covered by ice cloud (simply referred to as clear scene, liquid-cloudy scene, and ice-cloudy scene hereafter). We do not consider situations that are partly covered by both ice and liquid clouds. For each clear-cloudy combination, 20 randomly generated cloud fractions (10 for liquid clouds and 10 for ice clouds) were

applied by using IPA. Moreover, considering the situations that a cloud can be observed in a viewing angle but missed in another angle, because of parallax effect or cloud movement during the multi-angle data acquisition (Stap et al., 2016), 20% of pixels are assigned perturbed cloud fractions in different viewing angles during the IPA procedure. The cloud fraction angular perturbation range is restricted within both 0.2 absolute value and 100% of the cloud fraction itself, which corresponds best to the angular patterns seen in the PARASOL measurements. The full training set is randomly divided into two parts: about 7.5

million samples in the training set and 1.6 million in the test set.





**Table 1.** Details of the statistical distributions of the aerosol and cloud parameters used to generate the training datasets.

| parameter | min | max | mean | distribution |
|---|---|---|---|---|
| wind speed (m/s) | 0 | 87 | 7.52 | RemoTAP |
| chl-$\alpha$ concentration | 0 | 10 | 1.92 | RemoTAP |
| Li-sparse | 0 | 0.35 | 0.14 | RemoTAP |
| Ross-thick | 0 | 1.4 | 0.41 | RemoTAP |
| Maignan bpdf | 0 | 10 | 3.02 | RemoTAP |
| brdf scaling coefficient (443nm) | 0 | 0.40 | 0.06 | RemoTAP |
| brdf scaling coefficient (490nm) | 0 | 0.45 | 0.10 | RemoTAP |
| brdf scaling coefficient (565nm) | 0 | 0.50 | 0.17 | RemoTAP |
| brdf scaling coefficient (670nm) | 0 | 0.65 | 0.23 | RemoTAP |
| brdf scaling coefficient (865nm) | 0 | 0.80 | 0.33 | RemoTAP |
| brdf scaling coefficient (1020nm) | 0 | 0.90 | 0.37 | RemoTAP |
| effective radius of liquid cloud ($\mu$m) | 3 | 25 | 14 | uniform |
| effective variance of liquid cloud | 0.03 | 0.35 | 0.19 | uniform |
| cloud optical thickness of liquid cloud | 1 | 40 | 10.6 | log-uniform |
| cloud layer height of liquid cloud (km) | 1 | 10 | 5.5 | uniform |
| effective radius of ice cloud ($\mu$m) | 10 | 60 | 30 | uniform |
| cloud optical thickness of ice cloud | 1 | 100 | 21.5 | log-uniform |
| cloud layer height of ice cloud (km) | 2 | 17 | 9.5 | uniform |
| aspect ratio of ice cloud crystals | 0.179 | 5.592 | 1.57 | log-uniform |
| distortion of ice cloud crystals | 0.1 | 0.7 | 0.4 | uniform |
| aerosol effective radius of fine mode | 0.02 | 0.57 | 0.14 | RemoTAP |
| aerosol effective variance of fine mode | 0.01 | 0.8 | 0.20 | RemoTAP |
| aerosol optical thickness of fine mode | 0 | 4.58 | 0.67 | log-uniform |
| aerosol effective radius of dust mode | 0.7 | 6.12 | 1.89 | RemoTAP |
| aerosol effective variance of dust mode | 0.01 | 0.8 | 0.58 | RemoTAP |
| aerosol optical thickness of dust mode | 0 | 3.95 | 0.60 | log-uniform |
| aerosol effective radius of soluble mode | 0.7 | 6.12 | 3.24 | RemoTAP |
| aerosol effective variance of soluble mode | 0.01 | 0.8 | 0.59 | RemoTAP |
| aerosol optical thickness of soluble mode | 0 | 3.95 | 0.60 | log-uniform |
| cloud fraction | 0 | 1 | 0.46 | empirical |



Training NNs on the whole 7.5-million-sample training set requires considerably large computational time and memory. Therefore, instead of training one NN on the whole training set, we choose the "neural network ensemble" approach (Hansen and Salamon, 1990), which is to equally and randomly separate the whole training set into several parts (here we divide the set into 16 parts) and to train NNs on each part. The final output is defined as the average of outputs from all the NN ensembles:

$$f_{\text{final}} = S(\frac{1}{N_{\text{ens}}} \sum_{i}^{N_{\text{ens}}} T_i), \tag{3}$$

where $S$ is the Sigmoid function, $N_{\text{ens}}$ is the number of NN ensembles, $T_i$ is the direct output of the $i$-th NN ensemble (defined in Eq.(2)). As is already explained and tested in Di Noia et al. (2019), such an approach can achieve similar performance as training one NN on the full training set.

The measurement noise is modeled as a Gaussian random number with a zero mean and a standard deviation of 2% relative noise for intensity and 0.007 absolute noise for degree of linear polarization (DoLP). Such noise is added to the training set as a form of regularization, as explained in Bishop (1995).

The input variables for the NNs are the Intensity, DoLP at 14 viewing angles, and the corresponding viewing geometries (Solar Zenith Angle, Viewing Zenith Angle, Relative Azimuth Angle, and Scattering Angle). The intensity and DoLP, as a function of wavelength and viewing angle, are compressed using a principal component analysis (PCA) before the training. Twenty-five principal components are retained for radiance and 33 for DoLP, in order to compress the measurements, which are the same numbers as in Di Noia et al. (2015).

The NNs are implemented using PyTorch (version 1.11.0, https://pytorch.org/) and designed as Multi-Layer Perceptrons (MLPs). The training of the NNs utilizes the Backpropagation (BP) algorithm (Rumelhart et al., 1986) and batch training with a batch size of 12,000. The Adam optimizer (Kingma and Ba, 2014) is employed to minimize the RMSE (Root Mean Square Error) loss function. The neural network architecture consists of three hidden layers with 80 neurons each layer for over ocean scenes and 40 neurons for over land scenes.

## 3.2 Synthetic results

Before applying the NNs to real PARASOL measurements, it was tested on the 1.6-million-sample test set outside the training set. In Table 2, the results of the neural network cloud fraction retrieval on the test set for both the land and ocean schemes are presented, showcasing the performance metrics including bias, Mean Absolute Error (MAE), and Root Mean Square Error (RMSE).

**Table 2.** Bias, MAE and RMSE of cloud fraction on synthetic test set.

|       | bias    | MAE    | RMSE   |
|-------|---------|--------|--------|
| ocean | -0.0084 | 0.0427 | 0.0745 |
| land  | -0.0112 | 0.0477 | 0.0811 |





To derive a cloud mask from the cloud fraction predictions, we need to set thresholds for the cloud fraction to determine whether a pixel is cloudy or not. Therefore, to find a suitable threshold, the performance of the cloud mask should be assessed, and we define the following three criteria:

1. information loss, calculated as the fraction of clear pixels, that are wrongly flagged as cloudy, relative to all clear pixels. A low information loss is desired for not discarding too many clear pixels where aerosol information can be retrieved;

   2. effectiveness, calculated as the fraction of correctly-flagged cloudy pixels with respect to all cloudy pixels. The higher the effectiveness, the fewer aerosol retrievals are attempted on cloudy pixels;

   3. overall agreement, defined as the fraction of correctly-flagged pixels.

As an example, the performance of the corresponding cloud masks over land on the synthetic test set is summarized in Figure 1. Here we evaluate 2 cases, one with a very strict definition of truly cloud-free pixels (CF $< 10^{-4}$) and one somewhat looser definition (CF $< 0.01$). Then we evaluate the NN cloud mask for different thresholds of the retrieved CF. The overall agreement, effectiveness, and information loss decrease when thresholds for the retrieved cloud fraction (CF) are increased. This is because a looser cloud mask (a larger threshold) is less likely to incorrectly discard clear pixels, however, also a higher

chance of including cloudy pixels. Comparing the results for the very strict definition of cloud-free (CF $< 10^{-4}$) and the looser one (CF $< 0.01$), we see that the NN has difficulty in identifying scenes with CF $< 10^{-4}$, as the overall agreement drops by 15-20 percent points when using the stricter definition of cloud-free. The same conclusions also hold for the retrievals over ocean. Furthermore, we also obtain very similar performance on synthetic measurement data set created using an aerosol layer height that varies between 1 and 8 km. This demonstrates that the NN cloud fraction retrieval is not hampered by the fact that

it is trained with the assumption that aerosols are located in the lowest 1 km of the atmosphere.

For applying a cloud mask before further aerosol retrievals, an important aspect is the cloud mask has a low information loss (keeps more clear pixels) instead of a very high effectiveness (less redundant aerosol retrievals on cloudy pixels). Therefore, we suggest that a threshold of 0.05 for the retrieved cloud fraction should be good for aerosol retrieval.

## 4   Application on PARASOL data

The test results on the synthetic data set provide only an initial consistency check to conclude the algorithm works theoretically. To further evaluate the performance of the NNs, we apply it to PARASOL L1 data in 2008. The results are compared with MODIS for the full year and with the RemoTAP goodness-of-fit cloud mask for 2 days. In addition, the MODIS and NN cloud mask are applied to RemoTAP aerosol retrievals for the year 2008 and the remaining aerosol properties are validated with AERONET.

### 4.1   Global cloud fraction distribution in 2008

Figure 2 shows maps of the annual mean cloud fraction for both the PARASOL-NN retrieval and MODIS. Both MODIS and PARASOL-NN see high levels of annual cloud fraction average in regions around the equator and around the 60° S. In the





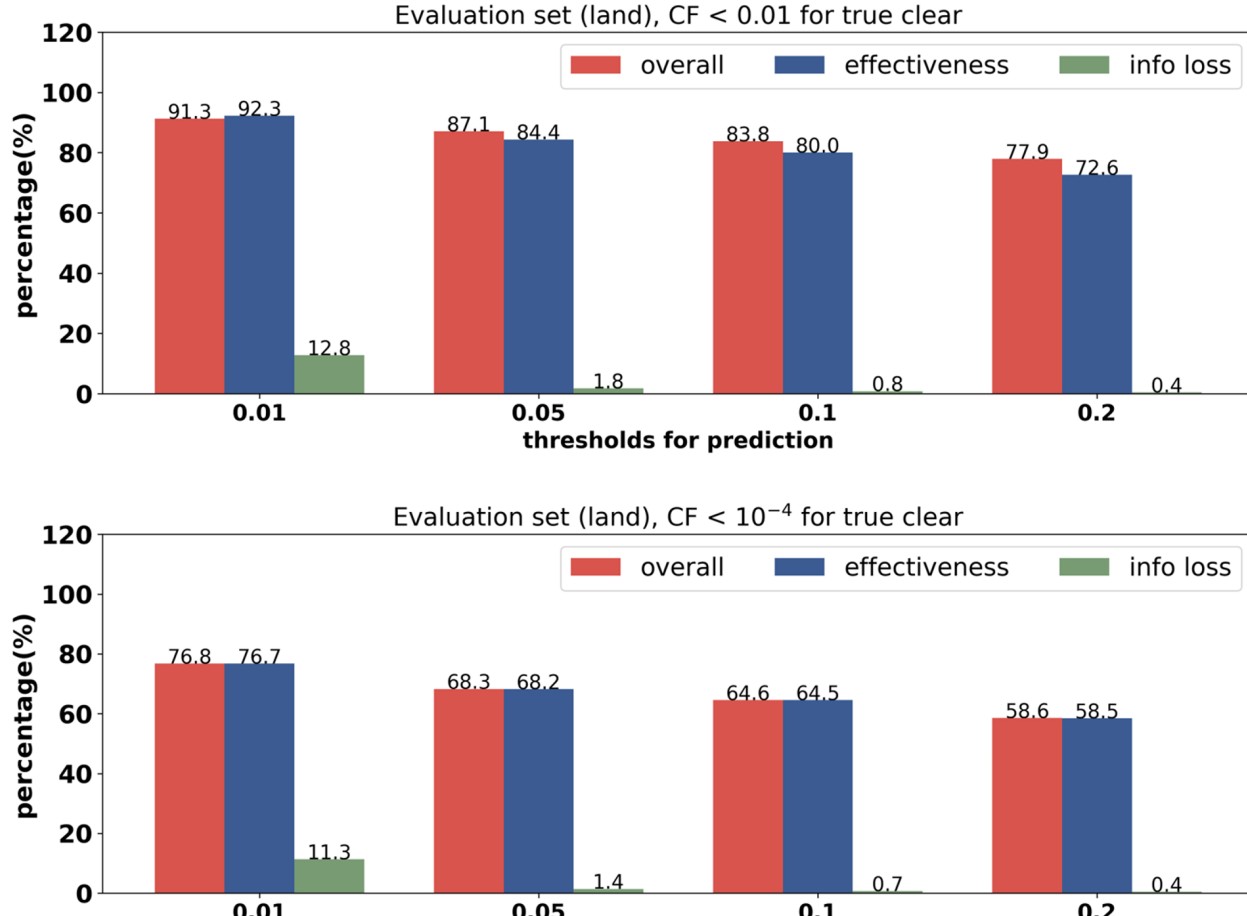

**Figure 1.** Performance of cloud mask over land on the synthetic test set for different thresholds. The upper panel uses "true" CF < 0.01 as truly clear while the lower panel uses CF < $10^{-4}$. The red bar stands for overall agreement. The blue bar represents the effectiveness of masking out cloud, and the green bar is the information loss.

desert areas (Sahara, Kalahari, Mojave, Atacama, and Great Victoria desert), MODIS and PARASOL-NN observe a quite low annual cloud fraction average. The latitudinal cloud fraction distribution also agrees well between PARASOL-NN and MODIS,
but PARASOL-NN retrieves on average a cloud fraction that is around 0.15 lower than MODIS.

Figure 3 shows the cloud fraction comparison between PARASOL-NN and MODIS for winter (Dec-Feb), spring (Mar-May), summer (June-Aug), and autumn (Sep-Nov). Similar to the full-year average, the NN retrieves a lower cloud fraction than MODIS for each season, but they both capture the same seasonal dependence of clouds in the tropics, subtropics, and also mid-latitudes. In tropical regions, near the equator, cloudiness tends to remain relatively high throughout the year. Subtropical
regions (20-30 degrees latitude), on the contrary, exhibit lower cloud fraction compared to the tropics. Seasonal variation





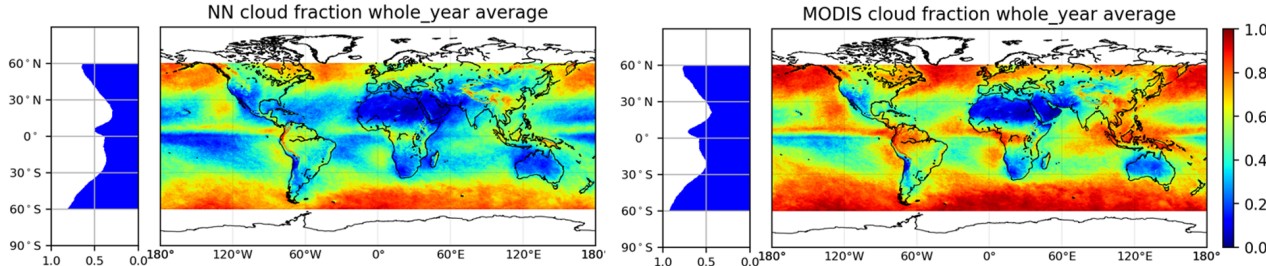

**Figure 2.** Whole year cloud fraction average (on each $\sim 6 \times 6$ km$^2$ PARASOL grid, shown in the colored world map) and latitudinal distribution (cloud fraction average along each longitude as a function of latitude, shown in the histogram). The left subplot shows the result from NN cloud fraction, and the right subplot is from MODIS cloud fraction.

of cloud fraction can be observed in mid-latitudes (30-60 degrees), but more significant variation appears in the monsoon-dominated regions. As an example, the Indian peninsula experiences distinct seasonal differences of a large cloud fraction in summer (due to the influence of the monsoon) and a low cloud fraction in winter. Both MODIS and PARASOL-NN agree well with those seasonal trends, but PARASOL-NN produces a lower cloud fraction.

Part of the reasons why the PARASOL-NN retrieved a lower cloud fraction than MODIS is that on the one hand, the PARASOL-NN seems to "miss" small sub-pixel clouds, but on the other hand, MODIS seems to sometimes misinterpret dust aerosol as a cloud, or overestimates the cloud fraction in some scenes. We illustrate this in Figure 4 with 2 examples where the cloud fractions from PARASOL-NN and MODIS are over-plotted on a background-corrected true reflectance image from MODIS (https://worldview.earthdata.nasa.gov/). Over the Red Sea (the upper row), MODIS interprets pixels with desert dust

as "confidently cloudy", while the PARASOL-NN can correctly see it as clear without missing the clouds in the area. This difference could be attributed to the contribution of the polarization information in the NN input, which is sensitive to particle size and shape, allowing it to effectively distinguish between clouds and thick aerosols (Hasekamp, 2010; Stap et al., 2014). Plots in the lower row show a partly cloudy area with small clouds (the ocean area to the south of Somalia). MODIS seems to see too large cloud fraction for parts of this scene while the NN predicts too low cloud fraction. A possible reason for the

cloud fraction underestimation by the NN is the movement of the clouds during the time of acquisition of all angles, leading to different cloud fractions to be sampled per viewing angle. Although we try to mitigate this effect in the NN training, some small moving clouds are still missed.

    Figure 5 shows a comparison between a strict MODIS (threshold 0.01) cloud mask and PARASOL-NN cloud mask for different cloud fraction thresholds. The overall agreement is about 80%. The effectiveness decreases somewhat towards the

larger cloud fraction threshold, which means that the percentage of filtered "true" (MODIS) cloudy pixels decreases with the threshold increases, whereas also the information loss decreases towards the larger cloud fraction threshold, which means that the percentage of wrongly flagged "true" (MODIS) clear pixels decreases with the increase of the threshold.

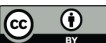

**Figure 3.** Seasonal cloud fraction average (on each $\sim 6 \times 6$ km$^2$ PARASOL grid, shown in the colored world map. From the upper line to the lower line, winter (Dec-Feb), spring (Mar-May), summer (June-Aug) and autumn (Sep-Nov)) and latitudinal distribution (cloud fraction average along each longitude as a function of latitude, shown in the histogram). The left subplot shows the result from NN cloud fraction, and the right subplot is from MODIS cloud fraction.





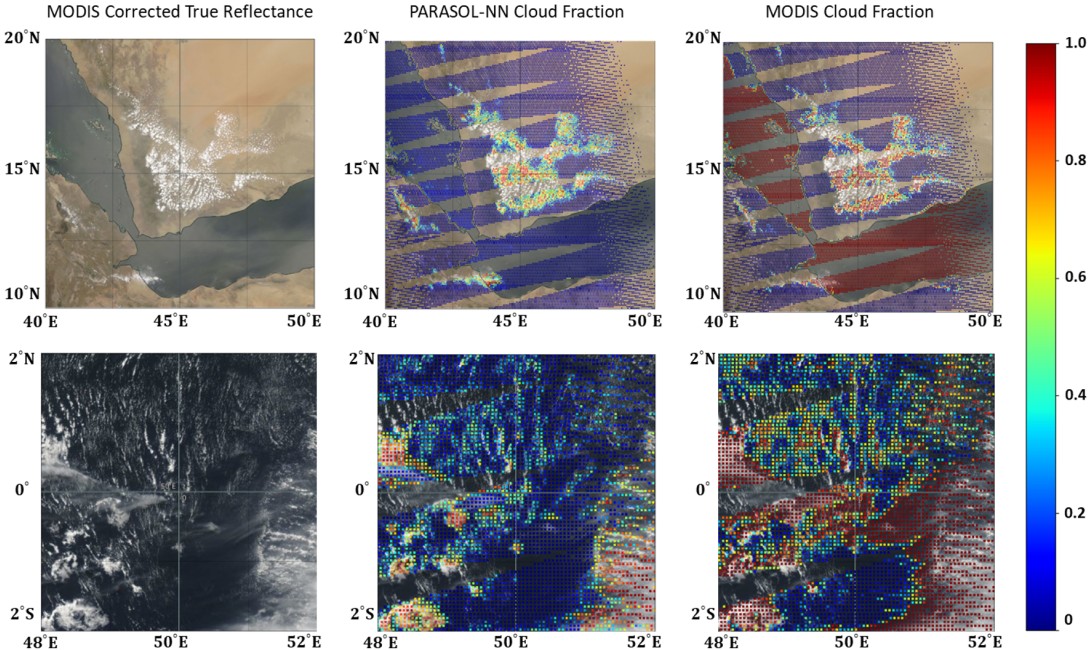

**Figure 4.** Cloud fraction around the Red Sea (the upper row) and to the south of Somalia (the lower row) on 1 July 2008. The left column is the MODIS corrected true reflectance from NASA worldview website. The middle column shows the NN cloud fracion and the right column is MODIS cloud fraction.

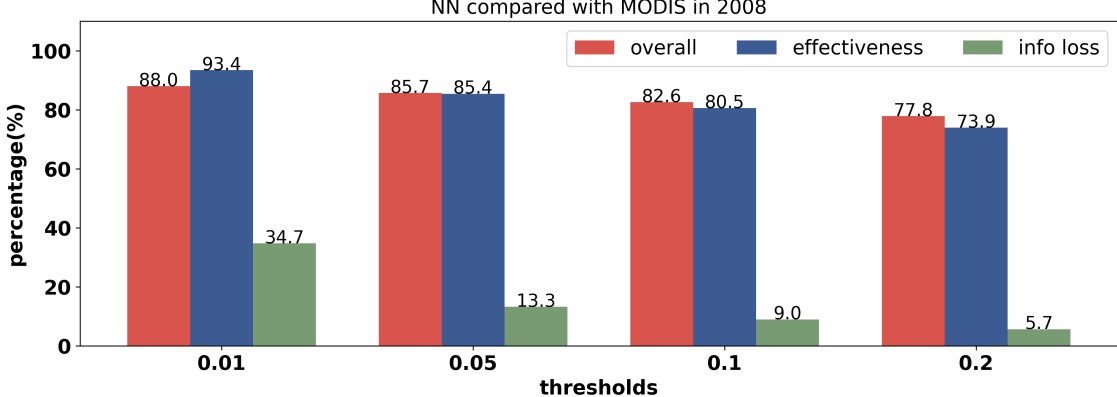

**Figure 5.** NN cloud mask compared with MODIS in the full year processing. The red bar stands for overall agreement. The blue bar represents the effectiveness of masking out cloud, and the green bar is the information loss.





## 4.2 Comparison to RemoTAP goodness-of-fit mask

As an additional assessment of the capability of the PARASOL-NN cloud mask, we perform RemoTAP aerosol retrievals on
all PARASOL pixels, both clear and cloudy, on two days (1 January and 1 July of 2008) to acquire a goodness-of-fit based
cloud mask. The high computational cost prevents us from a whole year retrieval without cloud screening, but these two days
contain both situations in summer and in winter, and thus should be reasonably representative.

The goodness-of-fit from a MAP aerosol retrieval is considered a good identifier of clear pixels, although it is important to
realize that the RemoTAP goodness-of-fit filter is not fully independent of the PARASOL-NN cloud mask because they both
rely on the same PARASOL level-1 measurements. The thought behind the goodness-of-fit cloud mask is that the forward
model in the aerosol retrieval scheme cannot reproduce cloud signals, which results in a large $\chi^2$. Based on previous works
(Stap et al., 2014, 2015), $\chi^2 < 5$ is a good threshold to identify cloud-free pixels.

The NN cloud fraction, MODIS cloud fraction and RemoTAP goodness-of-fit mask (blue area for $\chi^2 < 5$) are shown in
Figure 6. Both the NN and MODIS reveal a similar spatial pattern for the two days, while the NN seems more "blue" than
MODIS, which means the NN cloud fraction is overall smaller than for MODIS, as expected from the annual comparison.
Besides, both of them usually lead to cloud fractions either larger than 0.9 (mostly cloudy) or less than 0.1 (mostly clear). The
goodness-of-fit mask has a smaller cloud-free area than both NN and MODIS. Here, it should be noted that $\chi^2 > 5$ can also
happen due to other reasons than cloud contamination, i.e. pixels with large measurement errors, pixels for which the inversion
ends in a local minimum, or pixels where the aerosol and/or surface model in the RemoTAP forward model is not suited to
describe the actual atmospheric and/or surface conditions.

Figure 7 shows a comparison of the NN and MODIS cloud masks with the goodness-of-fit mask for four cloud fraction
thresholds. From this comparison, it follows that the information loss of the NN is considerably less than that of MODIS for
most thresholds. This indicates NN keeps more cloud-free pixels (as identified by the goodness-of-fit mask), and this feature
benefits a larger aerosol data coverage. The effectiveness of the NN, however, is lower than that of MODIS, leading to more
redundant aerosol retrievals that result in large $\chi^2$. Nevertheless, the effectiveness decrease is relatively small. As a balance of
effectiveness and information loss, the threshold of 0.05 is recommended, and it will be used to test the PARASOL-NN cloud
mask effect on retrieved aerosol properties in the next section.

## 4.3 Effect on retrieved aerosol properties

Undetected clouds can cause substantial biases in retrieved aerosol properties, such as an overestimation in AOD. However, a
too strict cloud mask leads to a reduced data coverage, especially in areas important to study aerosol-cloud interactions. In this
section, we assessed the quality of the retrieved aerosol properties after applying the PARASOL-NN or MODIS cloud mask.

We perform aerosol retrievals using RemoTAP for all AERONET-collocated PARASOL pixels in 2008 without applying any
prior cloud filter. Then we evaluate the retrieved aerosol properties against AERONET after applying different cloud masks.
We focus on the validation with AERONET on 3 aerosol properties:

1. the Aerosol Optical Depth (AOD, $\tau$) at 550nm;



**Figure 6.** NN cloud fraction (the first line), MODIS cloud fraction (the second line) and goodness-of-fit mask (the third line, blue area for $\chi^2 < 5$) on 1 Jan 2008 (the left column) and 1 July 2008 (the right column).

2. the Single Scattering Albedo (SSA) at 550nm;

3. the Angstrom Exponent (AE) for the wavelength pair 440-865nm, defined as

$$\alpha = -\frac{\log(\tau_{440}/\tau_{865})}{\log(440/865)}. \tag{4}$$

We evaluate the performance against the requirements formulated by the Global Climate Observing System (GCOS). For
AOD the GCOS requirement is that the AOD error should be smaller than 0.03 or 10% (whichever is greater). For AERONET
validation, this requirement has been modified in the Aerosol-CCI study (Popp et al., 2016) to 0.04 or 10% to also take into
account the uncertainty in AERONET AOD. For SSA the GCOS requirement is that the error should be smaller than 0.03. This





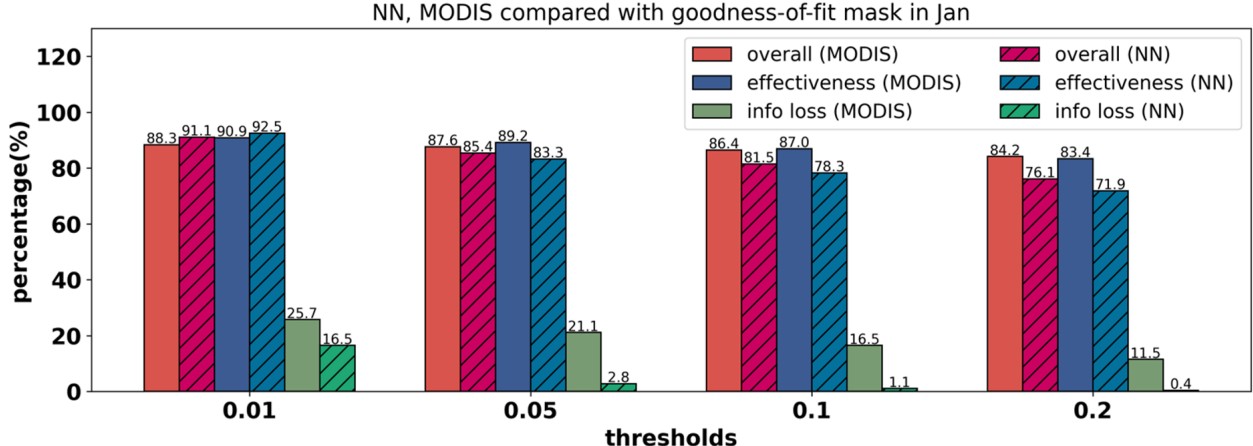

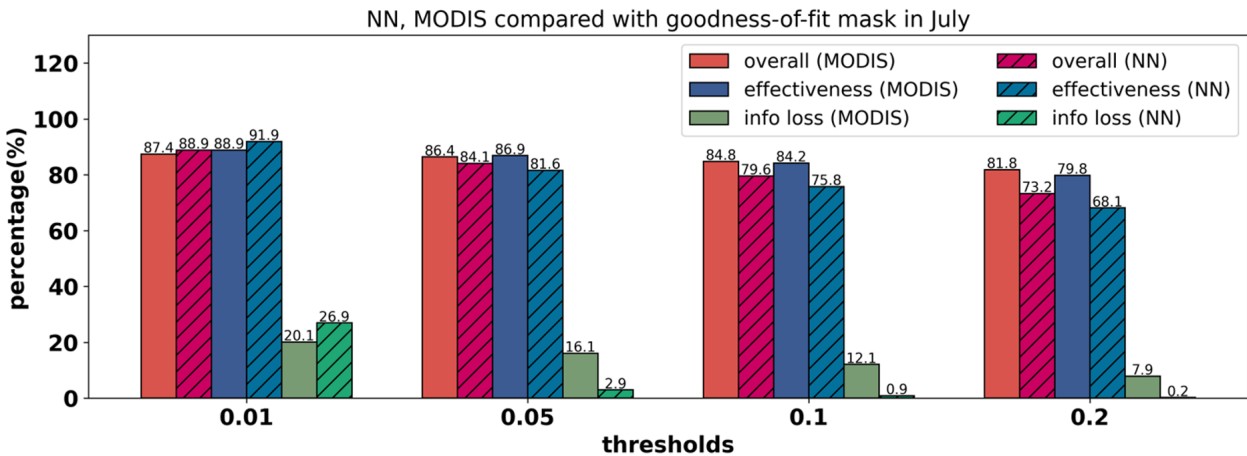

**Figure 7.** Comparison of the cloud masks with goodness-of-fit mask on 1 Jan 2008 and 1 July 2008. The upper panel shows the comparison on 1 Jan, and the lower panel is on 1 July. The comparison between NN cloud mask and goodness-of-fit mask is shown in shaded bars, while the comparison between MODIS and goodness-of-fit mask is in unshaded bars. The red bars stand for overall agreement. The blue bars represent the effectiveness of masking out cloud, and the green bars are the information loss.

requirement is not modified for AERONET evaluation given that the 0.03 requirement is considered already loose (Popp et al., 2016). For AE we use a requirement of 0.2 in line with what was used in the ESA HARPOL project (www.sron.nl/harpol).

Figure 8 shows scatter-plots (retrieved versus truth) for AOD, AE, and SSA after applying different cloud masks: PARASOL-NN with CF threshold 0.05 (the left column), MODIS with CF threshold of 0.05 (the middle column), and only the goodness-of-fit mask (the right column). It should be noted that the goodness-of-fit mask was also applied on top of the PARASOL-NN and MODIS cloud masks.



**Figure 8.** AERONET-collocated PARASOL retrievals compared with AERONET data (AOT, AE, and SSA, from the upper row to the lower row) after applying additional cloud mask (NN and MODIS, respectively, in the left column and the middle column, clear-cloudy threshold at 0.05) and only applying the goodness-of-fit mask ($\chi^2 < 5$ mask, the right column). In each subplot, vertical axis is the property retrieved from PARASOL data using RemoTAP, and the horizontal axis is AERONET data.





Figure 9 shows an overview of the percentage of pixels within the accuracy requirements, the RMSE, and the bias corre-
sponding to the 3 cloud masks. For both AOD and AE, applying a cloud mask (either MODIS or PARASOL-NN) can improve
the percentage of retrievals within the requirements, which suggests that either the goodness-of-fit filtered data still contain
some cloud-contaminated pixels or that the cloud filters screen out some other difficult pixels. Moreover, it is interesting to
see the bias of AE is negative for applying goodness-of-fit mask only and PARASOL-NN cloud mask, while that of MODIS
is positive instead. That is because MODIS filtered out a number of pixels with larger AE (see the plot in the second row and
second column) where more RemoTAP retrievals are smaller than AERONET data. Nevertheless, also for AE the biases for the
3 different cloud filters are small. The overall performance of the 3 different cloud masks is similar, but PARASOL-NN keeps
more pixels than MODIS (NN keeps 167136 pixels, MODIS keeps 160791 pixels and goodness-of-fit keeps 213578 pixels),
which is also consistent with the results in the two-day (1 Jan 2008 and 1 July 2008) comparison.

## 5   Conclusion

In this paper, we have presented an algorithm to filter clouds for aerosol retrievals from multi-angle, multi-wavelength polari-
metric measurements. The proposed approach is based on neural networks trained on synthetic measurements, where variations
in aerosol and surface properties as well as cloud macro- and microphysical properties, were taken into account. Separete NNs
have been trained for scenes over ocean and over land. With the NN we retrieve a cloud fraction at the native PARASOL Level
1B spatial resolution of $\sim 6 \times 6$ km$^2$, and this cloud fraction is subsequently used for cloud masking.
The neural network algorithm has been applied to process the entire PARASOL dataset for the year 2008, and a comparison
to the MODIS cloud fraction has been performed for seasonally and yearly averaged data. The comparison shows an overall
similarity in spatial patterns between PARASOL-NN and MODIS cloud fraction, while PARASOL-NN cloud fraction is around
0.15 smaller than MODIS. The lower cloud fraction in the NN can be partly explained by the fact that the NN sometimes
"misses" small sub-pixel clouds. Also, MODIS sometimes misinterprets dust aerosol as cloud while PARASOL-NN identifies
it correctly as cloud-free.
The comparison of the PARASOL-NN and MODIS cloud masks with the RemoTAP goodness-of-fit mask for two days (1
Jan and 1 July) shows the NN keeps more clear pixels for aerosol retrieval (low information loss) while the effectiveness in
filtering clouds is lower than MODIS. Nevertheless, the decrease in effectiveness is acceptable due to the substantially lower
information loss. Taking the balance of effectiveness and information loss, the NN cloud mask with CF threshold of 0.05 is
recommended for cloud screening before aerosol retrievals.
We also studied the effect of the cloud masks on retrieved aerosol properties. There is no significant difference in the ability
to remove the cloud-contaminated aerosol retrievals between PARASOL-NN and MODIS, while NN keeps more pixels than
MODIS.
The proposed algorithm could have a potential future application in analyzing data from recently developed multi-angle
polarimeters. By adjusting the instrument-specific factors, such as the number of viewing angles and spectral channels during
the neural network's training process, the algorithm can be adapted to work with instruments like the 3MI (Fougnie et al.,



**Figure 9.** Percentage of retrievals within requirements (GCOS/CCI requirement for AOD and SSA. HARPOL requirements for AE), RMSE and bias of the retrievals after applying additional cloud masks: NN cloud mask (red bar), MODIS cloud mask (blue bar) and only goodness-of-fit mask (green bar).



2018) on ESA/EUMETSAT Metop SG-A satellite, the Multi-Angle Polarimeter (MAP) instrument on the Copernicus CO2M mission (Spilling and Thales, 2021), as well as SPEXone (Hasekamp et al., 2019a) and the Hyper-Angular Rainbow Polarimeter (HARP-2), both of which are scheduled for launch in early 2024 on the NASA Plankton Aerosol, Cloud, ocean Ecosystem
(PACE) mission (Werdell et al., 2019).

### 5.0.1 Competing Interests

Some authors are members of the editorial board of journal Atmospheric Measurement Techniques.

### 5.0.2 Author contributions

ZY performed the experiments designed by OH, and BvD. GF performed RemoTAP aerosol retrievals. ZY wrote the first draft, which was further revised by OH, BvD and GF. HXL and JWE provided comments to the manuscript which improved the article, and provided advice on the research strategy.

*Acknowledgement.* ZY would like to thank the support from the China Scholarship Council (No. 202106220072). The authors also thank SURFsara, the Netherlands Supercomputing Centre in Amsterdam for providing the Spider cluster as the computing facility in the NN
training experiments (EINF-4298). Enrico Dammers (TNO) is acknowledged for providing helpful feedbacks on the research and article.





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
