# Peer review of "Cloud Detection from Multi-Angular Polarimetric Satellite Measurements using a Neural Network Ensemble Approach"

_Atmospheric Measurement Techniques, 2023_

## Referee Comment (RC1)

Review of "Cloud detection from multi-angular polarimetric satellite measurements using a neural-network ensemble approach" by Z. Yuan et al.

This manuscript describes a novel neural-network (NN) approach for cloud detection from on multi-angular polarimetric (MAP) measurements, which is here applied to the POLDER-3/PARASOL instrument. The main motivations put forward by the authors are mainly to use this product as a cloud mask for aerosol retrievals in order to avoid biases in case of multi-layer or coincident conditions within the same pixel (6°x6° for PARASOL). The NN approach is first presented and sensitivity analyses of some of its thresholds are discussed. Retrievals corresponding to one year of POLDER observations are shown and compared to MODIS. Comparisons to the more standard cloud masking approach "Goodness-of-fit" are also shown for two days. Finally, the impact of the choice of the mask (NN, MODIS or Goodness-of-fit) when doing aerosol retrievals is assessed by comparison to AERONET.

Advanced NN (or machine learning) approaches to detect and categorise clouds are increasingly used and excellent alternatives to replace or complement more traditional decision trees. Such an application for MAP measurement is in my knowledge new and is worth exploring. The method is timely considering the launch of PACE and 3MI in the very near-future. In this sense, this work is of scientific importance and fall within the scope of AMT. The overall NN approach seems sound and well constructed, although I must admit not being an expert.

The writing of the manuscript is fluent but some explanations (in particular of the method) lack clarity. Some figures are difficult to read, and overall lack a clear labelling of the panels; which makes it difficult sometimes to relate them to the text. The description of the figures is sometimes insufficient and overall there is a clear lack of quantitative analyses in the text. Some strong conclusions are taken without clear justification in my opinion. All these aspect unfortunately makes it difficult for the reader to properly understand the results of the new method and their added value to the other approaches illustrated here. Considering all these points, I advise for major revisions following the comments and suggestions below before considering this manuscript for publication.

**General comments to the authors**

1. A few comments regarding the method (described in section 3.1): Why are uniform distributions used to represent liquid and ice cloud effective radius, global dataset would clearly show a normal or log-normal distribution of these properties. What is the consequence of not considering the presence of both liquid and ice together on your method (and what is your reason for not considering them at all?) - these situations occur very often!
2. The conclusions of the comparisons to MODIS are too optimistic in my opinion. For instance l. 244-245, it is clear from the maps that the zonal fraction distribution does not agree very well. NN misses high cloud fractions in the tropics (along the ITCZ and in the warm pool region), see Fig 2. This seems to be related to convective clouds not being well identified by the NN approach. That can again be verified by looking at convective clouds over land during hemispheric summers not being identified by NN (e.g. above South America in the top row of Fig 3). Could you provide an explanation for this issue? And in any case it might be worth better discussing these limitations (that one and possible others) in your conclusion. You partly attribute the differences to the incorrect detection of aerosol as clouds by MODIS, which is true, but can't be the only reason. Also, the argument of cloud moving during the acquisition seems strange, how long does the acquisition take?
3. Have you considered providing the ice and liquid cloud fraction separately? This would make your method even more attractive and help to understand the performance of your method for instance by comparison to MODIS. One strong benefit of MAP measurements is the phase detection. It feels to me that this would be a small effort to add to your NN method considering the way it is constructed for a very high gain. In any case it is important to you characterise better what cloud type contribute to your uncertainties in CF estimates.
4. The main conclusion should be more clear. It seems to me that the NN approach leads to similar result to the goodness-of-fit method but that the latter is very computationally expensive. In that sense, a complete transition to the NN approach could be encouraged for future missions, is that correct?

**Specific comments**

1. There is no section describing the satellite data and its availability, as required in the AMT author guidelines. Additionally it would be useful to indicate in section 2.3 what version of the MODIS cloud mask is used.
2. Overall, there are large multi-panel figures that would require a labelling of the panels to be better understood, please add these and adjust the text discussions accordingly. Note that this is a requirement in AMT guidelines to authors.
3. l. 15: "Climate change, which refers to long-term changes in temperature and weather patterns" - this statement is too limiting, please revise it.
4. l. 23: Ice crystals are not directly formed from condensation nuclei, this is a bit misleading.
5. l. 57: Surely there are other reasons for retrievals not to fit (other non-retrieved parameters from the forward models). Could you be a bit more specific on how this method attributes the misfit to the presence of a cloud?
6. l. 78: "PARASOL contain unique sensitivity to clouds" - please be more specific.
7. Section 2.3: What version of the MODIS Cloud Mask have you used? In general, the paper lacks a clear description of the dataset that are used (including full name, DOI and access)
8. l. 217: "The higher the effectiveness, the fewer aerosol retrievals are attempted on cloudy pixels": please be more specific on what is meant here, is there more than one aerosol retrieval attempted on a cloudy pixel?
9. l. 255: What have you done to mitigate this effect? Be more specific.
10. Fig. 6: Very little can be seen on this figure in my opinion, I'd suggest removing it. The "NN seems more blue" (l. 284) is not quantitative enough for the standards here.
11. Fig. 8: This figure contains a lot of information and it is not clear what the point beyond the fact that there are little differences (at least notable in this figure) between POLDER using different masks and AERONET. Fig. 9 should be sufficient to make that point.
12. l. 320: "For both AOD and AE, applying a cloud mask (either MODIS or PARASOL-NN) can improve the percentage of retrievals within the requirements": it does not seem so clear from the figure, please be more quantitative in your analyses to justify this conclusion. There are some differences, but are they significant?
13. l. 351: Could you add some information on what you expect improvements would be for the method as applied to 3MI and PACE? Increased precision?

---

## Author Response (AR1)

**Response to reviewer 1**

We would like to thank the reviewer for his/her important comments and suggestions.

**1. A few comments regarding the method (described in section 3.1): Why are uniform distributions used to represent liquid and ice cloud effective radius, global dataset would clearly show a normal or log-normal distribution of these properties. What is the consequence of not considering the presence of both liquid and ice together on your method (and what is your reason for not considering them at all?) - these situations occur very often!**

*Response*: The purpose of this article is to develop a cloud masking algorithm for aerosol retrievals, which concerns mostly very small cloud fractions. We use uniform distributions for liquid / ice cloud effective radius in order to make sure that the NN has equal sensitivity to clouds with both small and large particles.    We added this statement to the manuscript at line 158 of the revised manuscript. Furthermore, Di Noia et al. (2019, https://doi.org/10.5194/amt-2018-345) has also used uniform distributions and demonstrated that this allows accurate retrieval of effective radius and variance.

The presence of both liquid and ice cloud together is indeed frequent. However, in case of pixels with small total cloud fractions (the focus of the present paper) these situations should occur less frequent.

To investigate whether considering scenes with both liquid and ice clouds on the cloud mask, we trained a new NN for a training set including also mixed liquid/ice scenes. From the comparison with aerosol retrieval goodness-of-fit mask (NN-chi2) and MODIS cloud mask on 1 July 2008, the performance isn't improved (the same conclusion also holds for retrieving liquid / ice cloud fraction separately).

Table 1. Performance on 1 July 2008 using training set with mixed-cloudy pixels.

**Comparison between cloud masks (global)**

| thresholds | NN-MODIS Similarity, effectiveness, info loss | NN-chi2 Similarity, effectiveness, info loss | MODIS-chi2 Similarity, effectiveness, info loss |
|---|---|---|---|
| 0.01 | 87.56%, 88.82%, 16.80% | 86.04%, 85.10%, 9.13% | 87.41%, 88.86%, 20.10% |
| 0.05 | 85.59%, 84.78%, 11.96% | 82.16%, 79.25%, 2.86% | 86.38%, 86.85%, 16.07% |
| 0.1 | 83.90%, 81.69%, 10.27% | 77.98%, 73.89%, 0.94% | 84.79%, 84.19%, 12.14% |
| 0.2 | 81.07%, 75.90%, 7.89% | 70.36%, 64.66%, 0.30% | 81.79%, 79.78%, 7.86% |

Experiment: mixed_cloudy_total on 20080701
Similarity: correct prediction / all
Effectiveness: successfully filtered clouds / all clouds
Info loss: wrongly flagged clear pixels / all clear pixels

Table 2. Performance on 1 July 2008 using training set without mixed-cloudy pixels.

**Comparison between cloud masks (global)**

| | NN-MODIS | NN-chi2 | MODIS-chi2 |
|---|---|---|---|
| thresholds | Similarity, effectiveness, info loss | Similarity, effectiveness, info loss | Similarity, effectiveness, info loss |
| 0.01 | 86.68%, 95.42%, 43.74% | 88.48%, 93.21%, 35.89% | 87.41%, 88.86%, 20.10% |
| 0.05 | 87.37%, 87.94%, 14.37% | 84.76%, 82.58%, 4.00% | 86.38%, 86.85%, 16.07% |
| 0.1 | 85.00%, 83.27%, 10.43% | 79.12%, 75.28%, 1.09% | 84.79%, 84.19%, 12.14% |
| 0.2 | 81.33%, 75.70%, 6.65% | 69.83%, 64.02%, 0.29% | 81.79%, 79.78%, 7.86% |

Experiment: I13Pf12 on 20080701
Similarity: correct prediction / all
Effectiveness: successfully filtered clouds / all clouds
Info loss: wrongly flagged clear pixels / all clear pixels

In the revised version, at line 182, we state that

"We do not consider situations that are partly covered by both ice and liquid clouds because experiments indicate that this does not improve the NN performance".

**2. The conclusions of the comparisons to MODIS are too optimistic in my opinion. For instance l.244-245, it is clear from the maps that the zonal fraction distribution does not agree very well. NN misses high cloud fractions in the tropics (along the ITCZ and in the warm pool region), see Fig 2. This seems to be related to convective clouds not being well identified by the NN approach. That can again be verified by looking at convective clouds over land during hemispheric summers not being identified by NN (e.g. above South America in the top row of Fig 3). Could you provide an explanation for this issue? And in any case it might be worth better discussing these limitations (that one and possible others) in your conclusion. You partly attribute the differences to the incorrect detection of aerosol as clouds by MODIS, which is true, but can't be the only reason. Also, the argument of cloud moving during the acquisition seems strange, how long does the acquisition take?**

*Response*: We rephrased the discussion of the global comparison to MODIS in the revised version, giving more attention to the limitation of the cloud fraction retrieved by the NN. We included a histogram of the cloud fractions retrieved by the NN and MODIS respectively. From this histogram it becomes clear that that the underestimation of the NN cloud fraction is mainly caused by the fact that the NN retrieves a cloud fraction close to 1 (0.95-1) for much fewer pixels (about 50%)   than MODIS (As shown below). For these pixels, the NN often retrieves cloud fraction in the range 0.8-0.95 instead.

[Figure]

Figure 2. Histogram of cloud fraction by MODIS and NN in the year 2008.

The NN also retrieves more cases with very low cloud fraction (0-0.05) which is caused by the fact that the NN misses some clouds and by the fact that MODIS confuses dust with clouds. The latter aspects are already discussed in detail in the paper. We have no indications that the differences are related to convective clouds.

The acquisition interval between MODIS and PARASOL can be several minutes, and it is different for each PARASOL viewing angle. In Figure 3 of this response we show the intensity at 760nm from one viewing angle and over-plot it on MODIS true reflectance. As is circled in the picture, some clouds indeed moved during the acquisition interval between MODIS and this PARASOL viewing angle.

[Figure]

Figure 3. PARASOL intensity at 670nm with MODIS true reflectance as background, 01 July 2008 near Somalia. The circled is moved cloud during the acquisition interval.

**3. Have you considered providing the ice and liquid cloud fraction separately? This would make your method even more attractive and help to understand the performance of your method for instance by comparison to MODIS. One strong benefit of MAP measurements is the phase detection. It feels to me that this would be a small effort to add to your NN method considering the way it is constructed for a very high gain. In any case it is important to you characterise better what cloud type contribute to your uncertainties in CF estimates.**

*Response*: To investigate whether considering scenes with both liquid and ice clouds on the cloud mask, we trained a new NN for a training set including also mixed liquid/ice scenes, where the NN outputs the liquid and ice CF separately.    However, the cloud mask did not improve. It is more challenging for the NN to retrieve two separate cloud fractions instead of one.

**4. The main conclusion should be more clear. It seems to me that the NN approach leads to similar result to the goodness-of- fit method but that the latter is very**

computationally expensive. In that sense, a complete transition to the NN approach could be encouraged for future missions, is that correct?

*Response*: The goodness-of-fit method is indeed effective but computationally expensive (because it will not be available before aerosol retrieval finished), so in the aerosol retrieval, we will first apply a cloud mask and then do aerosol retrieval on the remaining pixels. The goodness-of-fit mask is applied after retrievals finished to filter residual clouds. Instead of replacing goodness-of-fit mask completely, the conclusion of the article is that in the aerosol retrieval process, the NN cloud mask based on MAP measurements alone can replace the prior cloud mask provided by a cloud imager like MODIS. In the revised version the conclusions are reworded to make this more clear.

**Specific comments**

1. There is no section describing the satellite data and its availability, as required in the AMT author guidelines. Additionally it would be useful to indicate in section 2.3 what version of the MODIS cloud mask is used.

*Response*: We added the data availability section to the revised version. The question about the cloud mask version is answered below.

2. Overall, there are large multi-panel figures that would require a labelling of the panels to be better understood, please add these and adjust the text discussions accordingly. Note that this is a requirement in AMT guidelines to authors.

*Response*: We modified this in the revised version.

3. l. 15: "Climate change, which refers to long-term changes in temperature and weather patterns" - this statement is too limiting, please revise it.

*Response*: We removed this sentence in the revised version.

4. l. 23: Ice crystals are not directly formed from condensation nuclei, this is a bit misleading.

*Response*:     We rephrased it to: "Aerosols affect Earth's climate by scattering and absorbing radiation (aerosol-radiation interactions) and acting as condensation nuclei for cloud droplets and as ice nucleating particles to promote ice formation."

**5. l. 57: Surely there are other reasons for retrievals not to fit (other non-retrieved parameters from the forward models). Could you be a bit more specific on how this method attributes the misfit to the presence of a cloud?**

*Response*: We added the phrase: " …., because the forward model of a clear-sky aerosol retrieval cannot fit the characteristic spectral and angular signals caused by scattering on cloud particles (e.g. the cloud bow in polarization), as demonstrated by Stap et al. (2014, 2016, 2016).

**6. l. 78: "PARASOL contain unique sensitivity to clouds" - please be more specific.**

*Response*: The sentence "… that the MAP measurements of PARASOL contain unique sensitivity to clouds" means that MAP measurements can capture some polarimetric and/or angular features of clouds (such as cloudbow). We revised the paper as indicated in the response to the previous comment.

**7. Section 2.3: What version of the MODIS Cloud Mask have you used? In general, the paper lacks a clear description of the dataset that are used (including full name, DOI and access)**

*Response*: The version of MODIS is MYD35_L2, collection 6.1    (DOI: 10.5067/MODIS/MYD35_L2.061, last access: 5 Sept 2022). We added this in the revised version.

**8. l. 217: "The higher the effectiveness, the fewer aerosol retrievals are attempted on cloudy pixels": please be more specific on what is meant here, is there more than one aerosol retrieval attempted on a cloudy pixel?**

*Response*: After a cloud mask is applied, the aerosol retrievals will be applied on the clear-flagged pixels. When the "effectiveness" is higher, fewer actually cloudy pixels are wrongly flagged as clear, so there are fewer cases in which aerosol retrievals are applied on actually-cloudy pixels. We added this explanation to the revised version.

**9. l. 255: What have you done to mitigate this effect? Be more specific.**

*Response*: We applied different cloud fraction to each viewing angle during Independent Pixel Approximation in generating training set. The details can be found at Line 185 of the revised version.

**10. Fig. 6: Very little can be seen on this figure in my opinion, I'd suggest removing it. The "NN**

**seems more blue" (l. 284) is not quantitative enough for the standards here.**

*Response*: Based on the reviewer's suggestions, we remove the subfigures on 1 Jan 2008, but still keep the figures on 1 July 2008, because the figure 6 provides global comparison between NN / MODIS cloud fraction and the goodness-of-fit mask (where meaningful aerosol properties

can be retrieved). The information conveyed in the figure is that goodness-of-fit mask is much stricter to clear than NN / MODIS cloud mask (for some other reasons as listed in line 307 of revised version), so for validation of aerosol property quality, goodness-of-fit mask shall always be applied regardless of whether or which cloud mask is applied before aerosol retrieval. The quantitative comparison of them is shown in figure 7.

We rephrase "NN seems more blue" as "NN predicts more pixels with cloud fraction less than 0.1 (plotted in blue)" in the revised version l 304.

**11. Fig. 8: This figure contains a lot of information and it is not clear what the point beyond the fact that there are little differences (at least notable in this figure) between POLDER using**

**different masks and AERONET. Fig. 9 should be sufficient to make that point.**

*Response*: Based on the reviewer's suggestions, we removed the plots from the article.

**12. l. 320: "For both AOD and AE, applying a cloud mask (either MODIS or PARASOL-NN) can**

**improve the percentage of retrievals within the requirements": it does not seem so clear from the figure, please be more quantitative in your analyses to justify this conclusion. There are some differences, but are they significant?**

*Response*: Indeed the improvement is quite small, i.e. by about 2 percent-points as indicated by Fig. 10 a. Therefore we changed

  "… applying a cloud mask (either MODIS or PARASOL-NN) can improve the percentage of retrievals within the requirements, …"

to

"… applying a cloud mask (either MODIS or PARASOL-NN) can slightly improve the percentage of retrievals within the requirements, …" (adding the word 'slightly', l 337 of the revised version)

**13. l. 351: Could you add some information on what you expect improvements would be for the method as applied to 3MI and PACE? Increased precision?**

*Response*: We included this sentence to make the reader aware of the wider applicability of the approach (beyond PARASOL), but not necessarily to highlight improved performance of these instruments. Analysis of the sensitivity of the approach to instrument characteristics is subject to future work. We added this statement to the revised manuscript. (l 380, revised version)

**Response to reviewer 2**

We would like to thank the reviewer for his/her important comments and suggestions.

**Major comments:**

1. **I don't see much discussion on the accuracy of the cloud mask related to the measurement uncertainty itself, which, in my option, should be very important. The cloud fraction threshold which impact aerosol retrievals may also depend on the accuracy of the measurements.**

   *Response*:    It is hard to predict how the (required) accuracy of the cloud fraction is related to measurement accuracy. On the one hand, the measurement accuracy may affect the capability of the goodness-of-fit mask that is applied after the aerosol retrieval, i.e. the higher the measurement accuracy, the better the goodness-of-fit mask would work. On the other hand, for a high measurement accuracy, also the accuracy of retrieved aerosol properties is better and the effect of residual cloud may be more important relative to the aerosol retrieval accuracy. Another effect would be that the cloud fraction can be more accurately retrieved when the measurement accuracy is higher. Given the complexity of this discussion we believe it is a study on its own. We added in the conclusion the phrase:

   *"Application of the NN cloud screening approach to these new instruments will provide insight in the sensitivity of the approach to measurement uncertainty, number of viewing angles, and number of spectral bands."* Which provides an outlook to future studies.

2. **Almost all the measurements are used to derive a single cloud fraction. Cloud fraction seems easier to determine than cloud and aerosol microphysical properties, I wonder whether less measurements can still achieve reasonable performance. If less measurement can be used, maybe NN with smaller size and faster speed can be developed, or applied to fewer angles to gain more flexibility?**

   *Response*: The speed of the NN is not a concern at all for the current size of the NN (~12h to train and ~48h to retrieve the 2008 full year cloud fraction from PARASOL) so this would not be a reason to reduce the number of angles. We prefer to use all information available to perform the cloud screening. We do not see why there would be more flexibility when using less angles.

**Detailed comments:**

**P3, L62, "Given that the results for the cloudy pixels (~80% of all pixels)", 80% sounds too much, any reference?**

*Response*: In the context, cloudy pixels are defined in perspective of aerosol retrievals. Based on our previous global aerosol retrievals, around 80% of pixels are cloud-contaminated and thus not able to produce a reasonable aerosol retrieval result. Note that the percentage of cloudy pixels strongly depends on the sensor resolution. The 80% value is consistent with the analysis provided by Krijger et al. (2007; https://acp.copernicus.org/articles/7/2881/2007/acp-7-2881-2007.html) which is added to the paper as reference.

**P3, L70: "MODIS cloud mask, … is based on input signals from visible and infrared bands, which detect the high, spectrally flat reflectance and low brightness temperature feature of clouds.", The color on figure 4 over water is clearly not spectrally flat, just wonder why it is picked up by the MODIS cloud mask?**

*Response*: The MODIS cloud mask makes use of a number of thresholds and as a proxy of the spectrally flatness of the observations, the mask algorithm uses a ratio of reflectance at 865 and 670 nm. We do not have information which threshold combination has caused the dust scene to be flagged as cloudy by MODIS. Probably it is related to the spectral flatness (see above), the high signal level and a low brightness temperature (because of elevated dust). As a discussion on this bias of the MODIS cloud mask is beyond the scope of this paper, no changes were made to the paper related to this question.

**P3, L80, it may sound trivial, how the cloud fraction is defined? Cloud can cover partially in space or transparently over the pixel.**

*Response*: The cloud fraction is defined according to the independent pixel approximation (see Eq 1).

**P4, L106, why a minimum of 14 out of 16 angles are used? Can less angle be used?**

*Response*: Most PARASOL measurements have 14 angles. We added this as clarification to the revised version. We did not investigate the performance as number of viewing angles but for SPEXone (5 angles) we obtain similar performance as for PARASOL on synthetic measurements, but here the smaller number of angles may be compensated by a larger number of wavelengths.

**P4, L 106, What is the accuracy of the PARASOL measurements?**

*Response*: For the bands used in the study, we assumed intensity has a 2% relative noise and DoLP 0.007 absolute noise in the original version of the manuscript. Based on the questions of

the reviewer, we did a quick test on noise settings and found a variable relative noise between 1-3% for intensity and a 0.012 absolute noise for DoLP gave a better performance in term of effectiveness. Therefore, in the revised version, we use the latter noise settings.

**P5, L132, "The cloud fraction (referred to as MODIS cloud fraction hereafter) is calculated as the fraction of confidently- and uncertain-cloudy-flagged 1-km-resolution MODIS pixels within a 6km°ø6km PARASOL grid."**

**Should different weight be applied to the confidently cloudy and uncertain cloudy mask in calculating the cloud fraction? Since later the authors reported MODIS cloud fraction values are larger, would this be part of the reason?**

*Response*:   We tested using only confidently-cloudy flag as cloudy in collocating MODIS cloud fraction from MYD35 data, but no significant differences are observed. The reason is that there are usually not many MODIS (MYD35) uncertain pixels in a PARASOL grid, e.g. on 01 Jan 2008, 90% PARASOL grids have less than 5 MODIS uncertain flags, so we won't expected there are obvious differences between the two different definition of MODIS cloud fraction.

**P5, L149: one shape of ice crystal is used, does the shape matters? Would it possible to distinguish ice cloud or water cloud fraction maybe in future work?**

*Response*: The hexagonal ice crystals with varying aspect ratios and surface distortions represent the scattering properties of ice crystals with variable complex shapes. This is demonstrated in the reference given (van Diedenhoven et al. 2020) and references therein. We modified the sentence to mention that the hexagonal ice crystals are used as proxies for ice crystals with variable complex shapes. Furthermore, based on the results in the article, the current settings are sufficient for the main focus of cloud masking for aerosol retrievals. Distinguishing ice / liquid cloud fraction is one of our on-going work.

**P6, L178, "We do not consider situations that are partly covered by both ice and liquid clouds.". With the large pixel size of 6km x 6 km, there could be higher chances to observe partially covered cloud or cloud edge. I wonder whether the NN can be applied to every angle of the multi-angle observations, which may help detect partially covered cloud? There are some works which seems study such scenarios but with less efficiency (example: Gao et al, 2021: https://doi.org/10.3389/frsen.2021.757832), a more flexible NN cloud mask can be a remedy.**

*Response*: There are indeed situations where different angles see different cloud fractions because of inhomogeneity. Therefore, we include in our training set also samples where different angles have different cloud fraction. This increases the capability of the cloud screening (see l 185 of the revised version). We also added a reference to Gao et al, 2021 in the revised version.

**P7, Table 1, what does "RemoTAP" mean for distribution? You may already explain other-where, but it would be useful to include some information in the caption.**

*Response*: "RemoTAP" means the properties are randomly taken from RemoTAP global aerosol retrieval for the year 2008, as explained in line 159 of article. We added a short explanation in the caption of the table in the revised version.

**P8, L189, does smaller size of NN is needed when trained on parts of the total data?**

*Response*: We cannot give a universal answer to this question. Actually, the best size of NN depends more on the task itself (e.g. task complexity).

**P8, L205, "The neural network architecture consists of three hidden layers with 80 neurons each layer for over ocean scenes and 40 neurons for over land scenes."**

**I would expect cloud detection is harder over land due to its complexity, but here smaller NN is used over land comparing with the ocean one. Can you explain more?**

*Response*: The choice of NN is based on the performance on real measurements (with comparison to goodness-of-fit mask, i.e. experiment in section 4.2), which can reflect the generalization ability of the NN. Based on this performance we found smaller NN is better for retrievals over land than over ocean. An explanation can be that a larger NN usually has larger risk in overfitting data and thus struggling to generalize to new data. Therefore, sometimes it can be possible that a smaller NN behaves better for the more complicated over land case.

**P9, L233, "Therefore,we suggest that a threshold of 0.05 for the retrieved cloud fraction should be good for aerosol retrieval." I wonder whether the threshold should relate to the measurement uncertainty. Or how the measurement uncertainty relates to the accuracy of the cloud fraction accuracy?**

*Response*: See our response above to the general comment.

**P15, L309, "the AOD error should be smaller than 0.03 or 10%", can you confirm the definition of the error? As retrieval-truth or 1-sigma uncertainty etc?**

*Response*: AOD error is here defined as the absolute value of 'true-retrieved' . We add the explanation at l 335 of the revised version.

**Page 17, Figure 8, the title of the figure seems indicate goodness-of-fit has been applied for all three columns of the plots, but it seems not the case as indicated in the caption. Can you verify?**

*Response*: The goodness-of-fit mask is always applied on top of the NN and MODIS in this section of assessing their effect on aerosol retrievals. We revised the titles in the section to avoid this confusion. Figure 8 is moved to supplement.

**Page 19, Figure 9, first row, the maximum percentage is around 65%, which is roughly the percentage within 1-sigma of a Gaussian distribution. Is this the correct explanation?**

*Response*: Indeed if the 1-sigma uncertainty would be the same as the GCOS requirement this would result in 67% of the pixels within the requirement. This only is the case for SSA in our retrievals (although please note the number of validation points for SSA is low).